# Genetic diversity analysis of the invasive gall pest *Leptocybe invasa* (Hymenoptera: Apodemidae) from China

**Xin Peng**[1], **Hantang Wang**[1], **Chunhui Guo**[1], **Ping Hu**[1,2], **Lei Xu**[1], **Jing Zhou**[1], **Zhirou Ding**[1], **Zhende Yang**[1,2]*

**1** College of Forestry, Guangxi University, Nanning, Guangxi, China, **2** Guangxi Key Laboratory of Forest Ecology and Conservation, College of Forestry, Guangxi University, Nanning, Guangxi, China

* dzyang68@126.com

**Data Availability Statement:** All relevant data are within the paper and its Supporting Information files.

**Funding:** We thank the subsidization of the National Natural Science Foundation of China

## Abstract

*Leptocybe invasa* Fisher et LaSalle is a global invasive pest that seriously damages *Eucalyptus* plants. Studying the genetic diversity, genetic structure and introgression hybridization of *L. invasa* in China is of great significance for clarifying the breeding strategy, future invasion and diffusion trends of *L. invasa* in China and developing scientific prevention and control measures. Genetic diversity and phylogenetic analyses of 320 *L. invasa* female adults from 14 geographic populations in China were conducted using 10 polymorphic microsatellite loci (SSRs) and mitochondrial DNA cytochrome oxidase I gene sequences (COIs). (1) The Bayesian phylogenetic tree and haplotype network diagram showed that only haplotype Hap3 existed in *L. invasa* lineage B in China, while haplotypes Hap1 and Hap2 existed in lineage A, among which haplotype Hap2 was found for the first time. The nucleotide and haplotype diversities of lineage A were higher than those of lineage B. (2) The SSR genetic diversity of the Wuzhou Guangxi, Ganzhou Jiangxi and Panzhihua Sichuan populations was higher than that of the other 11 populations, and the SSR genetic diversity of lineage A was higher than that of lineage B. (3) The AMOVA analysis of mitochondrial COI data showed that 75.55% of the variation was among populations, and 99.86% of the variation was between lineages, while the AMOVA analysis of nuclear SSR data showed that 35.26% of the variation was among populations, and 47.04% of the variation was between lineages. There were obvious differences in the sources of variation between the COI and SSR data. (4) The optimal K value of COI and SSR data in structure analysis was 2, and PCoA analysis also divided the dataset into two obvious categories. The UPMGA phylogenetic tree based on SSR data clustered 14 geographic species into two groups. The results of genetic structure analysis supported the existence of two lineages, A and B, in China. (5) Structural analysis showed that there was obvious introgressive hybridization in Wuzhou Guangxi, Ganzhou Jiangxi, Panzhihua Sichuan and other populations. These results suggest that lineage introgressive hybridization has occurred in the *L. invasa* population in China. The introgressive hybridization degree and genetic diversity of lineage A are obviously higher than those of lineage B. Lineage introgressive hybridization may be the driving force for further *L. invasa* invasion and diffusion in China in the future.

(No.31971664, 31560212) and the Guangxi
Natural Science Foundation
(No.2018GXNSFAA294008,
2018GXNSFDA281004).

**Competing interests:** The authors have declared
that no competing interests exist.

## Introduction

With the development of international trade and economic globalization, biological invasion has become a serious concern within the international community. China is one of the countries most affected by invasive organisms in the world. At present, there are 560 confirmed invasive alien species; 125 are insect pests, 92 of which damage the agricultural ecosystem. The estimated annual economic loss due to alien invasive species is more than $18.9 billion [1].

*Leptocybe invasa* Fisher et LaSalle (Hymenoptera: Apodemidae) is an invasive gall pest that damages more than 30 species of *Eucalyptus* and mainly harms the branches and leaves of eucalyptus, causing typical lumplike galls on the veins, petioles and shoots of new leaves. Serious infection can lead to plant death and huge economic losses. *L. invasa* was first discovered in the Middle East and the Mediterranean in 2000 [2]. In 2014, it had spread to 29 countries on five continents [3]. In 2018, the number had increased dramatically to 45 countries [4]. *L. invasa* was first discovered in China in 2007 in Dongxing, Fangchenggang, Guangxi, which is near the border with Vietnam [5]. Currently, it has spread to Guangxi, Hainan, Guangdong, Fujian, Sichuan, Jiangxi, Yunnan, Taiwan and 8 other provinces [3,6–8]. *L. invasa* has become an important invasive forest pest of eucalyptus worldwide with unprecedented scale and speed of invasion and spread.

*L. invasa* has three lineages, A, B and C. Lineage A is mainly distributed in Europe, the Middle East, South America and most parts of Africa. Lineages A and B are distributed together in Laos, Thailand, Vietnam, South Africa and other places, while lineage C appears only in Australia [9,10]. Nugnes et al. first reported that *L. invasa* that invaded China belongs to lineage B [11]. However, Peng Xin et al. identified *L. invasa* from 14 geographical populations in China and found that there are both lineages A and B in China, with a ratio of approximately 1:2 [12].

The main reproductive method of *L. invasa* is parthenogenesis, in which the genes of the female offspring are the same as those of their mother [2]. Some of the symbiotic bacteria in *L. invasa*, such as *Rickettsia*, can induce parthenogenesis [11]. Due to its superior parthenogenesis, *L. invasa* can rapidly expand its population in invaded areas. In recent years, *L. invasa* males have been reported in Turkey (male/female ratio: 1:124) [13], India [14], China (male/female ratio: 1:126& 1:125) [15], Taiwan [16], Thailand [17] and other places successively. Liang Yiping's study found that the proportion of male *L. invasa* in Guangxi, China, was 18% - 48%, which was significantly higher than the proportions reported in other reports [18]. Yang Xianxian studied the reproductive biology of *L. invasa* and found that there is mating behavior between male and female *L. invasa* in China, and offspring can be produced through bisexual reproduction [19]. Through the analysis of genetic diversity and genetic structure, Dittrich-Schröder et al. found that *L. invasa* in the Vietnamese population had introgressive hybridization between lineages [9].

Introgression is a phenomenon in which genetic material from one species is transferred to another species through hybridization and repeated backcrossing, also called introgressive hybridization [20]. Hybridization occurs in approximately 10 percent of all animals and 25 percent of all plants in nature [21]. Hybridization is considered to be an important driving force for the adaptive evolution of species [22,23]. Introgression caused by hybridization has been reported in many animals, such as insects [24], mollusks [25], mammals [26], reptiles [27], amphibians [28], and fish [29]. In recent decades, with the frequent occurrence of invasive species introduction events, hybridization opportunities between invasive species and their relatives have increased, and introgressive hybridization progeny may be more invasive, leading to the extinction of local parent groups [30,31]. Introgressive hybridization easily occurs between lineages due to their close genetic relationships. Introgression leads to the

rapid improvement of genetic diversity, which increases the fitness of lineages in new environments and promotes the spread of lineages and population expansion [29,32,33]. The mixing of the *L. invasa* lineage may lead to introgression between lineages and the emergence of new genotypes that are more suitable for the environment [9]. In this paper, SSR and COI molecular markers were used to analyze the genetic diversity, genetic structure and introgressive hybridization of *L. invasa* in China, aiming to provide a basis for the prediction and scientific prevention and control of *L. invasa* invasion and spread trends in China.

## Material and methods

### Ethics statement

The sampling of living material involved in our experiments included *Leptocybe invasa*, associated with galls on *Eucalyptus*. All sampling locations are not privately owned or protected (coordinates in Table 1). Besides neither the *Eucalyptus* nor the *Leptocybe invasa* are endangered or otherwise protected, and therefore no specific permits were required for these locations and activities.

### Sample collection and DNA extraction

In this study, a total of 320 female adult *L. invasa* samples from 14 geographic populations in 6 provinces of China were collected from 2016–2020 (Table 1). All the samples were collected from DH201-2 (*Eucalyptus grandis* × *Eucalyptus tereticornis*) (Myrtales: Myrtaceae), except for GXNN2 (*Eucalyptus exserta*) (Myrtales: Myrtaceae).

Fresh eucalyptus shoots with galls were collected from each geographic population, and the branches were placed in a plastic bottle filled with water to maintain freshness and were then transferred into a sealed net cage (40 cm×40 cm×80 cm) at room temperature to keep the adults from escaping. The water in the plastic bottle was renewed daily until *L. invasa* adults emerged. Sexes were identified by morphological observation [3]. Samples of female adults from each geographic population were placed into tubes with anhydrous ethanol, labeled and stored in a refrigerator at -80˚C. To avoid contamination, only single geographical population

**Table 1. Sampling site information of 14 geographic populations of *L. invasa* in China.**

| Pop | Sample sites | Longitude and latitude | Elevation (m) | Data | Haplotype | GeneBank accession numbers | N |
|---|---|---|---|---|---|---|---|
| SCDY | Deyang, Sichuan | (104˚17′E, 31˚10′N) | 512.3 | 2019–12 | Hap1 | MZ378835-MZ378866 | 32 |
| SCPZH | Panzhihua, Sichuan | (101˚75′ E, 26˚49′ N) | 1137.2 | 2018–07 | Hap1, Hap3 | MZ379018-MZ379049 | 32 |
| JXGZ | Ganzhou, Jiangxi | (114˚43′ E, 25˚36′ N) | 195.6 | 2016–07 | Hap1, Hap3 | MZ378903-MZ378934 | 32 |
| FJSM | Sanming, Fujian | (117˚57′ E, 26˚30′ N) | 177.6 | 2019–09 | Hap3 | MZ378867-MZ378898 | 32 |
| HNDZ | Danzhou, Hainan | (108˚56′ E, 18˚09′ N) | 24.6 | 2017–07 | Hap3 | MZ378899-MZ378902 | 4 |
| YNKM | Kunming, Yunnan | (103˚10′ E, 26˚6′ N) | 1846.5 | 2019–07 | Hap3 | MZ378935-MZ378937 | 3 |
| GXFCG1 | Fangchenggang1, Gaungxi | (108˚1′E, 22˚14′ N) | 250.0 | 2019–07 | Hap1, Hap2, Hap3 | MZ379056-MZ379087 | 32 |
| GXFXG2 | Fangchenggang2, Gaungxi | (107˚47′E, 21˚57′N) | 220.0 | 2019–10 | Hap1, Hap3 | MZ379088-MZ379119 | 32 |
| GXNN1 | Nanning1, Guangxi | (108˚17′ E, 22˚51′ N) | 143.3 | 2019–06 | Hap1, Hap3 | MZ378986-MZ379017 | 32 |
| GXNN2 | Nanning2, Guangxi | (108˚17′ E, 22˚50′N) | 156.4 | 2019–06 | Hap1, Hap3 | MZ378954-MZ378985 | 32 |
| GXWZ | Wuzhou, Guangxi | (111˚03′ E, 23˚03′ N) | 76.2 | 2016–07 | Hap1, Hap3 | MZ379120-MZ379151 | 32 |
| GXLB | Laibin, Guangxi | (108˚24′ E,23˚16′ N) | 361.3 | 2020–06 | Hap3 | MZ378938-MZ378953 | 16 |
| GXYL | Yulin, Guangxi | (110˚01′ E,22˚30′ N) | 150.46 | 2019–08 | Hap3 | MZ379152-MZ379154 | 3 |
| GXQZ | Qinzhou, Guangxi | (109˚12′ E, 21˚52′ N) | 45.26 | 2020–07 | Hap3 | MZ379050-MZ379055 | 6 |

Note: Pop: Population code; N: Number of samples.

*L. invasa* samples were cultivated each time, and the remaining branches were treated in time after the culture. Thirty-two samples were randomly selected from the samples collected from each geographic population for subsequent molecular analysis; some populations with insufficient samples required the use of the entire sample.

An animal tissue DNA extraction kit (Beijing Tsingke Biological Technology) was used to extract the total genomic DNA of single *L. invasa* female adults according to the instructions. A Nanodrop 2000 spectrophotometer was used to measure the quality and concentration of DNA at 260 nm/280 nm and 260 nm/230 nm ratios, and the qualified DNA was stored in a freezer at -20°C for later use.

## Mitochondrial COI gene sequence download

A total of 320 COI sequences of 14 geographical populations in China (MZ378835–MZ379154) [12] and 511 COI sequences of 18 countries on 5 continents (MH093001–MH093481, JQ289999–JQ290005, KP233972–KP233993, KP233954) [9,11] were downloaded from the GenBank molecular database.

## Microsatellite amplification and capillary electrophoresis genotyping

References were made to the polymorphic SSR primers of *L. invasa* developed by Dittrich-Schröder et al. [9] and Peng Xin et al. [12] (Table 2). The 5' end of the forward primer was labeled with four kinds of fluorescence (FAM, ROX, HEX and TAMAR) (Beijing Tsingke Biological Technology). The sample DNA used was consistent with mitochondrial sequencing. The PCR system was 25 μL, including 12.5 μL of 2×T5 Super PCR Mix (Beijing Tsingke

**Table 2. Information on 10 polymorphic SSR primers in *L. invasa*.**

| ID | F/R | Primers(5'-3') | FL | Reapeat motifs | $T_a$ | NA | Alleles size | PIC |
|---|---|---|---|---|---|---|---|---|
| c120771 | F | AGCCAAAAGGGGTTTGTTCT | FAM | (AG)6 | 56 | 2 | 248, 252 | 0.331 |
| | R | ACTCAGCAACAGGTGTCACG | | | | | | |
| c124062 | F | CGTCTGTTCAGTCCTCCTCC | FAM | (CG)7 | 58 | 2 | 145, 155 | 0.330 |
| | R | CATTGCAAGCTACAGTCCGA | | | | | | |
| c120888 | F | GCGCGCGTCTATATACTTCC | HEX | (CT)6 | 57 | 2 | 233, 239 | 0.115 |
| | R | TACGCGCACGAGTTGTATGT | | | | | | |
| c121460 | F | CGCTCTCACGAGGAGAGACT | ROX | (ATA)7 | 58 | 3 | 196, 199, 217 | 0.136 |
| | R | ACATCCGCGACAACTTCTCT | | | | | | |
| c69914 | F | ATCGAATGGCCGTATTTCAA | FAM | (TA)10 | 54 | 4 | 198, 200, 202, 204 | 0.483 |
| | R | CTTGCGGAGAAATCAAGGAG | | | | | | |
| c121749 | F | GTATACGAGGGGGAGGGAAA | ROX | (CAG)6 | 58 | 2 | 249, 258 | 0.320 |
| | R | ACAGTTGCTGCTGTACGTGG | | | | | | |
| c127471 | F | ATGTCGAAGGGCAATTTCTG | TAMRA | (TGT)7 | 56 | 2 | 220, 229 | 0.247 |
| | R | ACTCGGAATTCAATCAACGC | | | | | | |
| c123946 | F | GTGTCGATTGGCGCTATTGT | HEX | (GCT)5 | 56 | 4 | 237, 240, 246, 249 | 0.383 |
| | R | CGTGTGTGAGAGTGCGAAAT | | | | | | |
| LiSS5 | F | TCGTGTTTACCACCTGACCA | ROX | (AGC)9 | 56 | 2 | 351, 354 | 0.150 |
| | R | AGAGTGCTCAGGCTCGACAT | | | | | | |
| LiSS13 | F | TGGTACAAATCCCGTCTATGG | FAM | (ACGC)7 | 54 | 2 | 141, 149 | 0.131 |
| | R | CGCAACGGTACAGAAATTCA | | | | | | |

Note: F: Forward primer sequence; R: Reverse primer sequence; FL: Fluorescent label; $T_a$: Optimal annealing temperature; NA: Number of alleles; PIC: Polymorphic information content.

Biological Technology), 1.0 μL each of forward fluorescent primer and reverse primer (diluted to 10 μm), 1.0 μL of DNA template, and 9.5 μL of ddH$_2$O. The PCR procedure was predenaturation at 98˚C for 3 min, 35 cycles of denaturation at 98˚C for 10 s, annealing at a suitable temperature for 10 s, extension at 72˚C for 10 s, and extension at 72˚C for 2 min. PCR products were sent to Beijing Tsingke Biological Technology for capillary electrophoresis genotyping using a QIAXCEL nucleic acid analysis system (QIAGEN, Irvine, USA) (S1 Fig).

## Bayesian phylogenetic tree and haplotype network diagram construction

A total of 831 COI sequence data downloaded from the GenBank molecular database were input into Notepad++ v6.5 software and saved in FASTA format (S1 Table). The COI sequences were aligned by PhyloSuite v1.1.1 [34]. The optimal model of the Bayesian tree was found through the Model Finder program in PhyloSuite v1.1.1 software, and the optimal model was used to construct the Bayesian phylogenetic tree. The generation number was set as 100 million, and 3 replicates were run. The *L. invasa* COI gene sequence haplotype network was constructed using Popart v1.7 software (http://popart.otago.ac.nz).

## Genetic diversity analysis

Genetic diversity analysis based on SSR data: The GeneAlex v6.5 plug-in [35] in Excel was used to process the capillary electrophoresis results. Genetic diversity analysis software was used to analyze the genetic diversity of (1) different geographic populations (N = 320), (2) lineage A (N = 104), and (3) lineage B (N = 216). Arlequin v3.5 [36] was used to calculate genetic diversity indices, including the total number of alleles, average number of alleles, number of polymorphic loci, average observed heterozygosity, average expected heterozygosity and average genetic diversity. Arlequin v3.5 [36] was used to perform molecular analysis of variance (AMOVA) to quantitatively analyze the genetic variation within and among populations and between and within lineages. Genepop v4.2 [37] was used to detect linkage disequilibrium, and a significant correction of linkage disequilibrium was achieved through the "sequential Bonferroni correction" [38] (S2 Table). Genepop v4.2 [37] was used to detect Hardy-Weinberg equilibrium at each site (S3 Table), and Cervus v3.0 [39] software was used to calculate polymorphic information content (PIC) at each locus.

Genetic diversity analysis based on COI data: The haplotype and nucleotide diversities of different lineages and all samples were calculated using DNAsp v6 [40]. Arlequin v3.5 [36] was used to analyze the molecular variance of *L. invasa* within and among populations within and between lineages in China.

## Genetic structure analysis

Population v1.2.32 software (http://bioinformatics.org/project/?group_id=84) was used to construct a UPMGA population evolutionary tree based on SSR data. Structure version 2.3 (http://pritch.bsd.uchicago.edu/structure.html), which applies a Bayesian clustering method, was used to indicate the number of populations or clusters (K) based on the genetic identity of the individuals and without prior geographic information. An admixture model with correlated allele frequencies was implemented because it was more likely to detect subtle population structures. Tests included 20 repetitions for a range of K-values from 1 to 10. For the Markov chain Monte Carlo (MCMC) procedure, the 'burn in' period was set to 100000 iterations followed by 100000 samples. Structure Harvester [41] is used to determine the likelihood value and ΔK [42] changes to verify the optimal K value, which represents the most likely clustering of individuals. The results were visualized using an online tool (http://pophelper.com/). Principal coordinate analysis (PCoA) was performed on SSR and COI data using GenAlEx v6.5 [35].

## Results

### Bayesian phylogenetic tree and haplotype network diagram

A Bayesian phylogenetic tree and haplotype network were constructed using 831 605-bp COI gene sequences downloaded from GenBank. The Bayesian tree and haplotype network diagram showed that there are three lineages of *L. invasa*: A (Hap1-Hap2), B (Hap3-Hap25), and C (Hap26-Hap27). Haplotypes Hap4-Hap25 and Hap26-Hap27 were found only in Australia, while haplotypes Hap3 and Hap1 were widely distributed in countries other than Australia where *L. invasa* was invaded except Australia. Two lineages, lineage A and lineage B, exist in China, including Hap1, Hap2 and Hap3, among which Hap2 was first found in the GXFCG1 population in China (Figs 1 and 2).

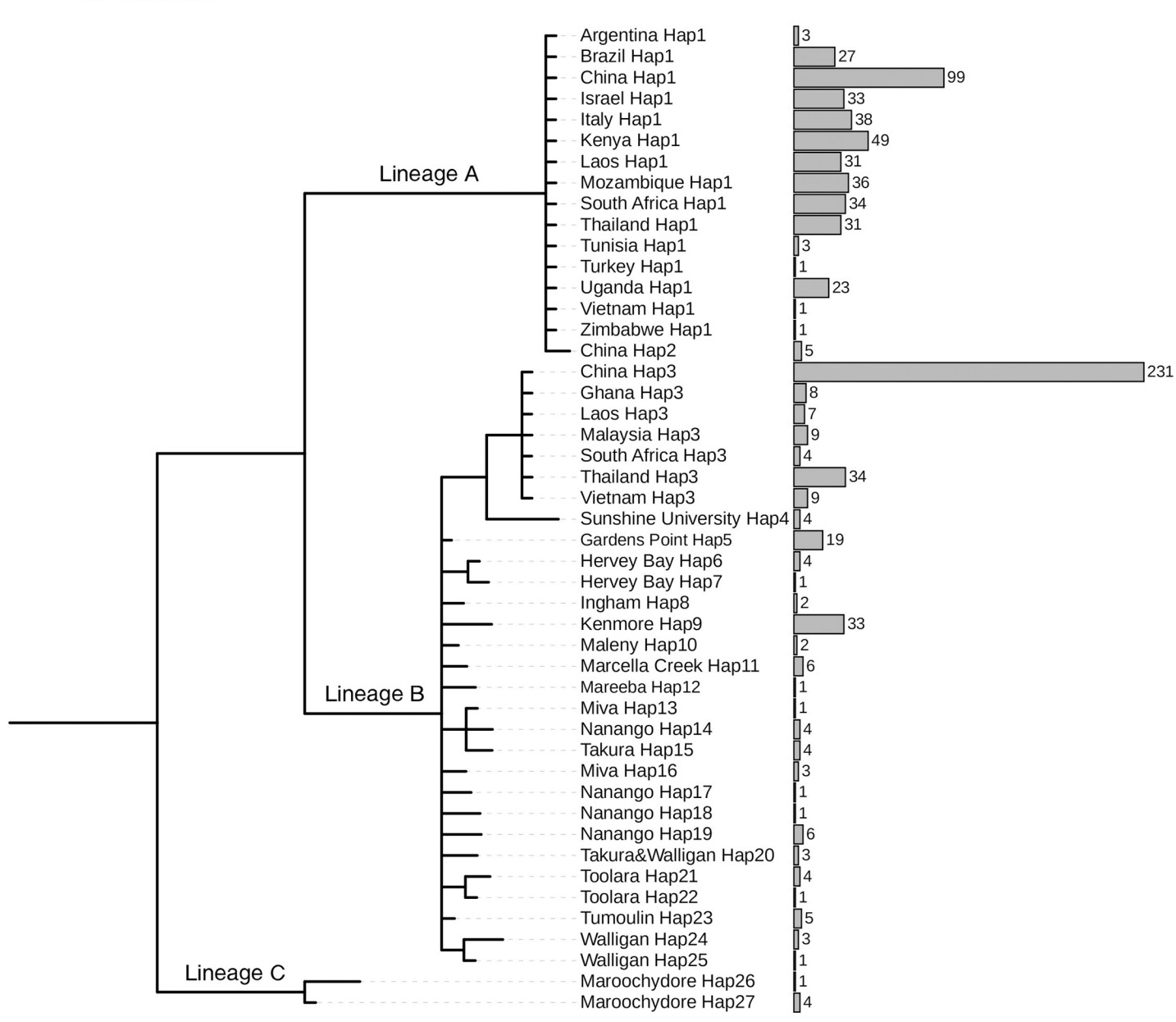

**Fig 1. Bayesian phylogenetic tree of *L. invasa* COI gene sequence.**

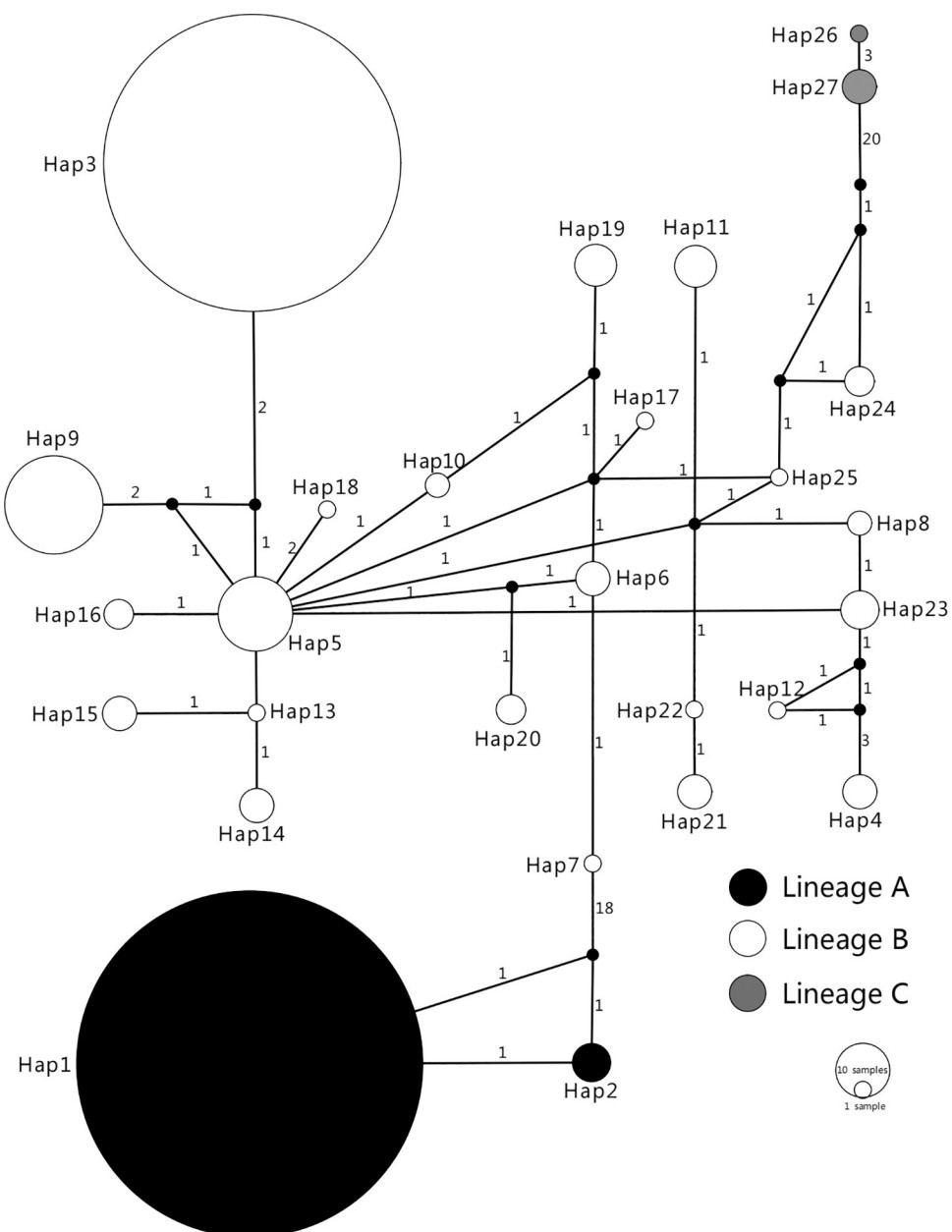

**Fig 2. Network diagram of *L. invasa* COI gene sequence.**

### Genetic diversity

The genetic diversity analysis of COI data showed (Table 3) that the base and nucleotide diversities of *L. invasa* lineage A were significantly lower than those of lineages B and C. There was no base mutation in the COI gene sequence of lineage B in China, and lineage A had only one base mutation site. Nucleotide and haplotype diversities at the lineage level were both at low levels, indicating that *L. invasa* in China has recently experienced population expansion. AMOVA analysis (Table 4) showed that the genetic variation among and within populations accounted for 75.55% and 24.45%, respectively, while the genetic variation between and within lineages accounted for 99.86% and 0.14%, respectively. Thus, genetic variation was concentrated among populations and lineages.

**Table 3. Genetic diversity analysis of available *L. invasa* mitochondrial COI gene sequences.**

| Region | Lineage type | Sample size | Number of polymorphic site | Number of haplotypes | Haplotype Diversity (Hd) | Nucleotide diversity (Pi) |
|--------|--------------|-------------|----------------------------|----------------------|--------------------------|---------------------------|
| Global | Lineage A | 415 | 1 | 2 | 0.024 | 0.00004 |
| | Lineage B | 411 | 25 | 23 | 0.451 | 0.00338 |
| | Lineage C | 5 | 3 | 2 | 0.400 | 0.00198 |
| | Total | 831 | 51 | 27 | 0.623 | 0.01891 |
| China | Lineage A | 104 | 1 | 2 | 0.092 | 0.00015 |
| | Lineage B | 216 | 0 | 1 | 0.000 | 0.00000 |
| | Total | 320 | 21 | 3 | 0.450 | 0.01529 |

Diversity analysis of SSR data showed (Table 5) that at the population level, JXGZ had the most alleles, GXWZ had the highest observed heterozygosity and expected heterozygosity, and the genetic diversity of the GXWZ, JXGZ and SCPZH populations was significantly higher than that of the other 11 populations. At the lineage level, the total numbers of alleles of lineages A and B were similar, but the observed heterozygosity, expected heterozygosity, number of polymorphic sites and average genetic diversity of lineage A were significantly higher than those of lineage B. AMOVA analysis (Table 4) showed that the genetic variation among and within populations accounted for 38.61% and 61.39%, respectively, while the genetic variation between and within lineages accounted for 48.86% and 51.14%, respectively. Compared with the COI gene sequence, the genetic variation of SSR was concentrated within the populations and lineages.

## Genetic structure

In the UPMGA population evolutionary tree constructed based on SSR data (Fig 3), the GXFCG1, JXGZ, GXWZ and SCPZH populations were clustered into one branch, and the other populations were clustered into another branch, which was consistent with the results of structure analysis.

Principal coordinate analysis (PCoA) of COI data showed that lineages A and B were clearly separated, and coordinates 1 and 2 represented 99.65% and 0.35% variation percentages, respectively (Fig 4). The principal coordinate analysis of SSR data showed that there was a mixing region between lineages A and B, and coordinates 1 and 2 represented 44.32% and 12.26%

**Table 4. AMOVA analysis of *L. invasa* based on SSR and COI data in China.**

| Molecular marker type | Source of variation | Degree of freedom | Sum of squares | Variance components | Percentage of variation | Fixation index |
|-----------------------|---------------------|-------------------|----------------|---------------------|-------------------------|----------------|
| COI | Among Populations | 13 | 1104.341 | 3.75132 Va | 75.55 | Fst: 0.7555* |
| | Within Populations | 306 | 371.406 | 1.21375Vb | 24.45 | |
| | Total | 319 | 1475.747 | 4.96506 | 100 | |
| | Among Lineage | 1 | 1470.987 | 10.47701 Va | 99.86 | Fst: 0.9986* |
| | Within Lineage | 318 | 4.760 | 0.01497 Vb | 0.14 | |
| | Total | 319 | 1475.747 | 10.49198 | 100 | |
| SSR | Among Populations | 13 | 341.644 | 0.57427 Va | 35.26 | Fst:0.3526* |
| | Within Populations | 626 | 648.548 | 1.05432 Vb | 64.74 | |
| | Total | 639 | 990.193 | 1.62858 | 100 | |
| | Among Lineage | 1 | 275.610 | 1.01117 Va | 47.04 | Fst: 0.4704* |
| | Within Lineage | 638 | 714.583 | 1.13845Vb | 52.96 | |
| | Total | 639 | 990.193 | 2.14962 | 100 | |

**Table 5. SSR genetic diversity analysis of *L. invasa* in China.**

| POP | N | NA | NAM | NP | HO | HOP | HE | GD |
|---|---|---|---|---|---|---|---|---|
| SCDY | 32 | 14 | 1.400±0.306 | 2 | 0.163±0.112 | 0.813±0.188 | 0.115±0.078 | 0.117±0.079 |
| JXGZ | 32 | 22 | 2.200±0.133 | 10 | 0.362±0.102 | 0.362±0.102 | 0.356±0.065 | 0.363±0.066 |
| GXFCG1 | 32 | 19 | 1.900±0.180 | 8 | 0.297±0.123 | 0.037±0.143 | 0.241±0.068 | 0.245±0.069 |
| GXFCG2 | 32 | 17 | 1.700±0.153 | 7 | 0.122±0.094 | 0.174±0.132 | 0.108±0.052 | 0.109±0.053 |
| GXNN1 | 32 | 14 | 1.400±0.163 | 4 | 0.159±0.105 | 0.398±0.227 | 0.102±0.061 | 0.104±0.062 |
| FJSM | 32 | 12 | 1.200±0.133 | 2 | 0.134±0.102 | 0.672±0.328 | 0.078±0.055 | 0.080±0.056 |
| GXNN2 | 32 | 17 | 1.700±0.153 | 7 | 0.222±0.115 | 0.317±0.153 | 0.174±0.063 | 0.177±0.064 |
| SCPZH | 32 | 20 | 2.000±0.149 | 9 | 0.273±0.092 | 0.303±0.097 | 0.331±0.063 | 0.336±0.064 |
| GXWZ | 32 | 21 | 2.100±0.100 | 10 | 0.375±0.090 | 0.375±0.090 | 0.421±0.027 | 0.429±0.027 |
| GXLB | 16 | 15 | 1.500±0.224 | 4 | 0.125±0.099 | 0.313±0.232 | 0.114±0.058 | 0.118±0.060 |
| GXQZ | 6 | 15 | 1.500±0.224 | 4 | 0.150±0.101 | 0.375±0.219 | 0.154±0.074 | 0.168±0.080 |
| HNDZ | 4 | 14 | 1.400±0.221 | 3 | 0.150±0.100 | 0.500±0.250 | 0.163±0.084 | 0.186±0.096 |
| GXYL | 3 | 13 | 1.300±0.153 | 3 | 0.200±0.113 | 0.667±0.192 | 0.122±0.065 | 0.147±0.077 |
| YNKM | 3 | 14 | 1.400±0.221 | 3 | 0.167±0.090 | 0.556±0.111 | 0.144±0.074 | 0.173±0.088 |
| Mean | | | 1.621±0.054 | 5.429±0.789 | 0.207±0.027 | 0.443±0.176 | 0.187±0.019 | 0.197±0.020 |
| Lineage A | 104 | 23 | 2.300±0.213 | 10 | 0.354±0.071 | 0.354±0.071 | 0.400±0.039 | 0.403±0.040 |
| Lineage B | 216 | 21 | 2.100±0.277 | 8 | 0.166±0.100 | 0.208±0.122 | 0.151±0.062 | 0.151±0.063 |

Note: POP: Population; N: Sample size; NA: Total number of alleles; NAM: Mean number of alleles by Locus; NP: Number of polymorphic loci; HO: Mean observed heterozygosity; HOP: Mean observed heterozygosity for polymorphic loci; HE: Mean expected heterozygosity (±SD); GD: Mean genetic diversity of loci.

variation percentages, respectively (Fig 4), indicating the presence of gene introgression between lineages.

The optimal K value of structure analysis based on COI data was 2. Black and yellow represent the samples of lineages A and B, respectively. Lineages A and B appeared simultaneously in 7 populations, namely, GXFCG1, JXGZ, GXWZ, SCPZH, GXFCG2, GXNN1 and GXNN2 (Fig 5).

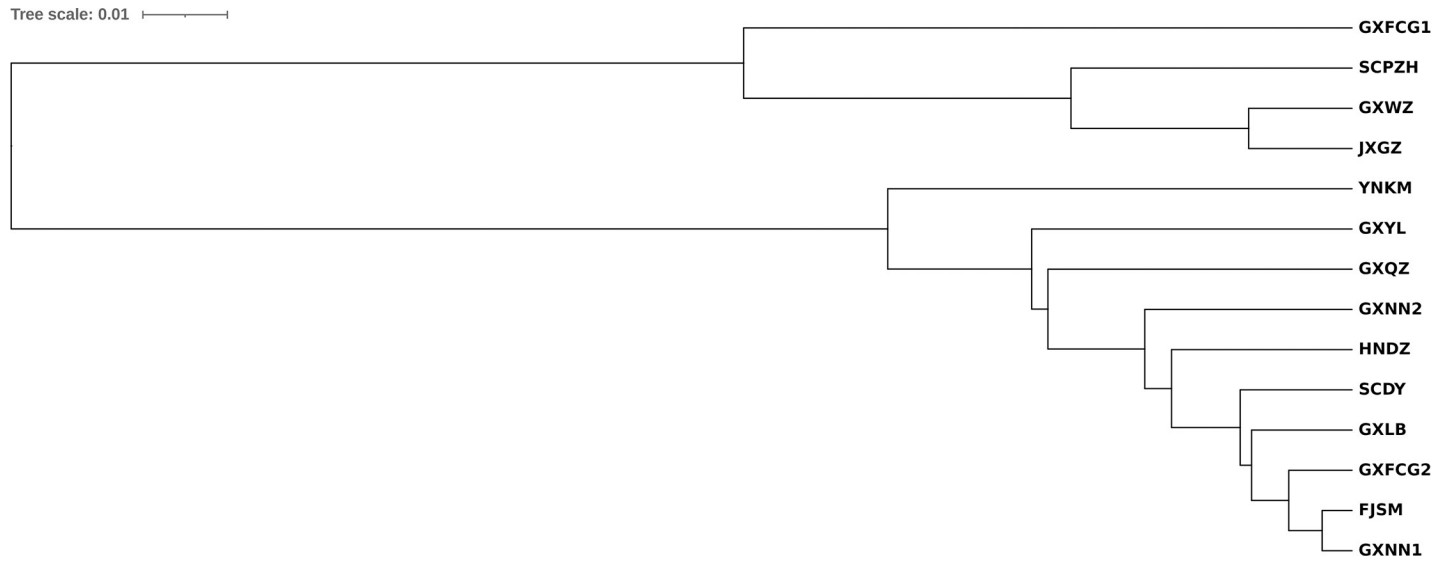

**Fig 3. UPMGA phylogenetic tree constructed from 14 populations of *L. invasa* based on SSR data in China.**

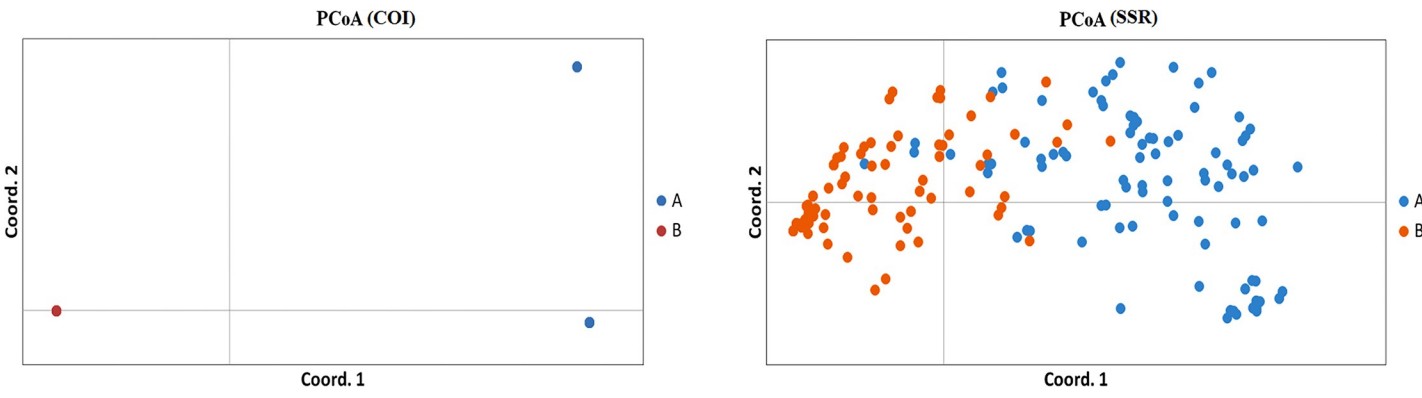

**Fig 4. PCoA of *L. invasa* based on the COI and SSR data in China.**

The optimal K value of the structure analysis based on SSR data was also 2. Black and yellow represent the genes of lineage A and lineage B, respectively. Lineage mixing appeared in some of the samples, indicating that the samples were introgression hybrids. The samples of the GXFCG1, JXGZ, GXWZ and SCPZH populations were dominated by the genes of lineage A, while the other populations were dominated by the genes of lineage B. Different degrees of introgressive hybridization appeared in the JXGZ, GXWZ, SCPZH, GXNN2, GXQZ, GXYL, GXLB and HNDZ populations. *L. invasa* in the GXWZ and SCPZH populations was almost replaced by their introgressed hybrids (Fig 5).

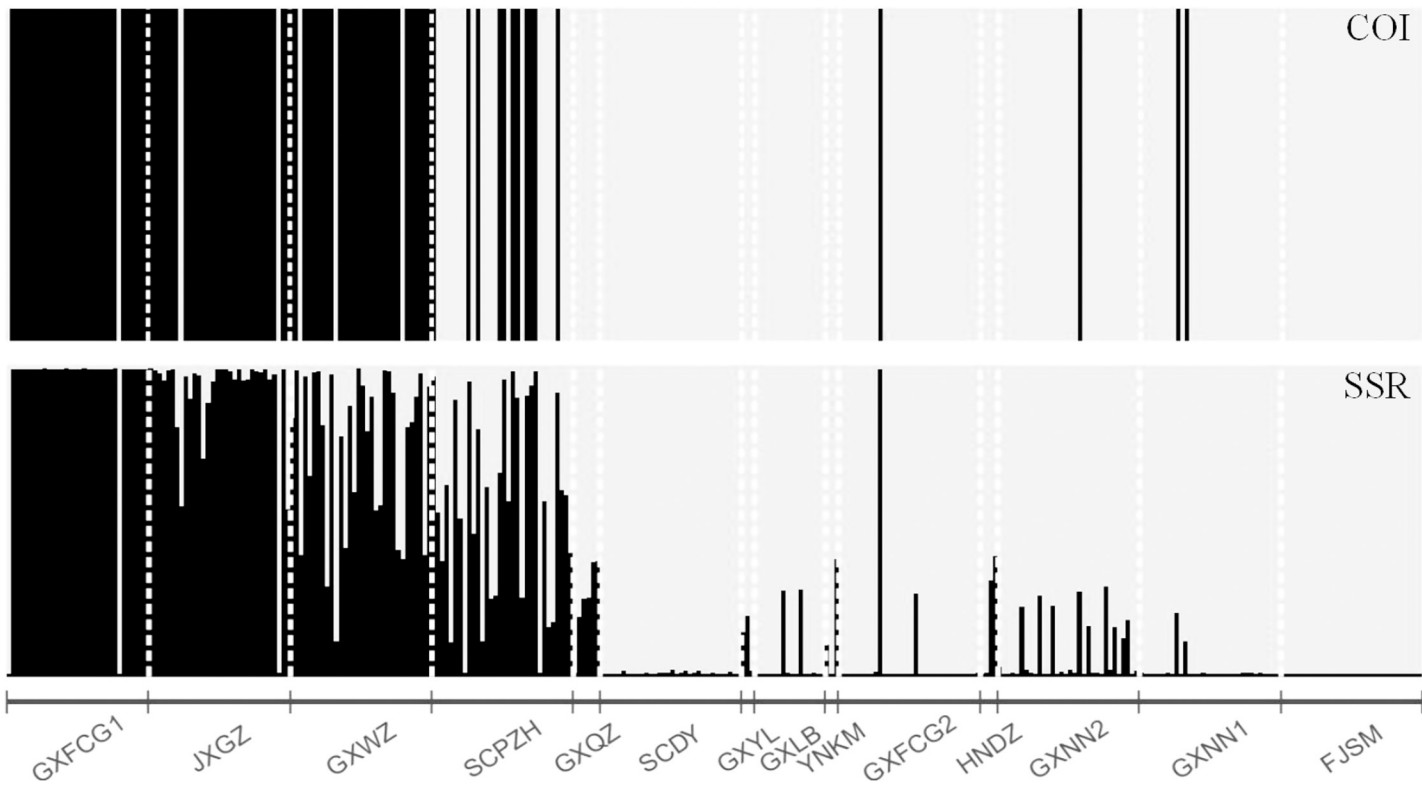

**Fig 5. STRUCTURE analysis of *L. invasa* based on COI and SSR data in China.**

## Discussion

We captured *L. invasa* female adult samples from 14 geographic populations in 6 provinces of China, covering most of the *L. invasa* hazard areas in China. In particular, Guangxi Province, where the *L. invasa* outbreak first occurred, has set up multiple sampling sites to attempt to identify its possible transmission route. Because all susceptible varieties of eucalyptus have been replaced with highly resistant varieties in Guangdong Province, we did not collect samples of *L. invasa* from the Guangdong population, which indicated that the comprehensive promotion of resistant varieties has a very good effect on the control of *L. invasa*. There are two lineages of *L. invasa* in China: lineages A and B [12], which are consistent with the lineages in other countries in the world where *L. invasa* has invaded. However, a new haplotype, HAP2, was found in lineage A of the Chinese population by Bayesian clustering analysis, which is of guiding significance for the traceability of lineage A. The lineage structure of *L. invasa* in China is similar to that in Southeast Asia, which indicates that *L. invasa* in China probably originated from Southeast Asia.

When selecting appropriate pest control methods, it is necessary to consider the genetic diversity, genetic structure and gene flow of the pest population, which can help us predict the future distribution of invasive pests, the possibility of hybridization between different lineages, and the ability to overcome the resistance of different host plants. Mixtures between closely related species are of particular concern, as introgression within a mixed population can rapidly increase the genetic diversity within the population, thus enabling pests to invade previously unsuitable environments or to overcome control measures [43,44]. The progeny of parthenogenesis are genetically the same as the mother, and the genetic diversity of its population is mainly determined by a few founders. The genotypes of its progeny are highly consistent, which is the root cause of the low polymorphism of *L. invasa* SSR loci [9]. The genetic structure analysis of *L. invasa* specimens from different geographical populations in China that was performed using structure and PCoA showed that there was obvious introgressive hybridization in China, which was similar to the genetic structure analysis results of Dittrich-Schröder et al. in Laos [9]. Introgressive hybridization among lineages significantly increased the genetic diversity in some geographic populations, such as Wuzhou in Guangxi, Panzhihua in Sichuan and Ganzhou in Jiangxi. By AMOVA, the variation percentage within lineages based on COI data (99.84%) was significantly higher than that based on SSR data (44.86%). This phenomenon of inconsistency in the source of nucleoplasmic gene variation also proved that introgressive hybridization occurred between lineages. According to the genetic structure of the species, there were heterozygous samples in both lineages A and B, which indicated that the introgression between lineages of *L. invasa* may be bidirectional. Le et al. speculated that *L. invasa* lineage B might be more inclined to produce male offspring [4]. Therefore, in the process of introgressive hybridization, lineage B is more likely to infiltrate genes into lineage A through sexual reproduction, which makes the genetic diversity and introgressive hybridization degree of lineage A significantly higher than that of lineage B. In addition, survival pressure may also be an important factor affecting the *L. invasa* reproductive strategy. When the food supply is insufficient, *L. invasa* will increase male differentiation, thus reducing the reproductive ability and the number of offspring. For example, the proportion of males collected from *E. exserta* was 1.7–1.8 times that of DH201-2 (*E. grandis* ×*E. tereticornis*) [18]. In this study, the introgressive hybridization degree of the Guangxi Nanning 2 population was significantly higher than that of the Guangxi Nanning 1 population, which also indicated that the host plants would affect the reproductive strategy of *L. invasa*. In recent years, China has vigorously promoted resistant varieties of eucalyptus, resulting in the lack of suitable hosts for *L.*

*invasa* and insufficient food supply, leading to the transformation of the *L. invasa* reproductive strategy and more frequent sexual reproduction in the population.

The emergence of introgressive hybridization in lineages has sounded the alarm for the control of *L. invasa* in China. Hybrids have been widely distributed in several populations in China. Among them, *L. invasa* populations in Guangxi Wuzhou and Sichuan Panzhihua were almost replaced by their introgressed hybrid offspring, which indicates that there is obvious heterosis in the *L. invasa* population. For most organisms, heterosis can only be maintained for one generation and is difficult to inherit steadily. However, *L. invasa* can rapidly expand the number of offspring of dominant hybrids through parthenogenesis, thus breaking the reproductive barrier to avoid inbreeding decline and maintaining heterosis in the population for a long time. Hybrids can form large populations independent of their parents in a short period of time. Once a hybrid is able to adapt to the new environment, it can invalidate the previously dependent resistant varieties and specific natural enemies and, in a short time, outbreaks of insect pests can occur once again. There are a large number of clonal eucalyptus forests in China, which will become a huge hidden problem for *L. invasa* control. In particular, the remaining stands of susceptible eucalyptus species, including DH201-1, DH201-2 (*E. grandis* × *E. tereticornis*) and *E. exserta* margin. *L. invasa*, can quickly establish large populations of susceptible host plants. The aggregation of a large number of progenies increases the chance of gene exchange between lineages, and susceptible host plants accelerate the generation replacement and evolution rate of pests. Susceptible eucalyptus species can not only serve as the source of *L. invasa*'s continuous spread but also become a hotbed for *L. invasa* to adapt to the environment through genetic variation. With the intensification of *L. invasa* lineage mixing, gene penetration between lineages will become a trend in the future. *L. invasa* flexibly applies two reproductive strategies to adapt to the environment, which will increase the difficulty of its prevention and control.

## Conclusion

Introgressive hybridization between *L. invasa* lineages A and B has occurred in many geographical populations in China. The genetic diversity and introgression degree of *L. invasa* lineage A were significantly higher than those of lineage B. Lineage introgressive hybridization may be the driving force for *L. invasa* to invade and spread in China in the future.

## Supporting information

**S1 Fig. *L. invasa* SSR capillary electrophoresis peak.**
(DOCX)

**S1 Table. *L. invasa* sample GenBank accession numbers used to construct the Bayesian phylogenetic tree and haplotype network diagram.**
(DOCX)

**S2 Table. Calibration results of linkage imbalance detection of 10 SSR primers of *L. invasa*.**
(DOCX)

**S3 Table. Hardy–Weinberg equilibrium test for 14 geographic populations of *L. invasa*.**
(DOCX)

## Author Contributions

**Funding acquisition:** Zhende Yang.

**Software:** Hantang Wang, Chunhui Guo, Ping Hu, Lei Xu, Jing Zhou, Zhirou Ding.

**Writing – original draft:** Xin Peng.

**Writing – review & editing:** Xin Peng, Zhende Yang.

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
