## [Decision Letter · Decision Letter 0]

8 Sep 2021

PONE-D-21-20406Genetic diversity analysis of invasive gall-pest Leptocybe invasa (Hymenoptera: Apodemidae) from ChinaPLOS ONE

Dear Dr. Yang,

Thank you for submitting your manuscript to PLOS ONE. After careful consideration, we feel that it has merit but does not fully meet PLOS ONE’s publication criteria as it currently stands. Therefore, we invite you to submit a revised version of the manuscript that addresses the points raised during the review process.

We look forward to receiving your revised manuscript.

Kind regards,

Bi-Song Yue, Ph.D

Academic Editor

PLOS ONE

“We thank Lei Xu and Hantang Wang for insect collection. And thank the subsidization of the National Natural Science Foundation of China (No.31971664, 31560212) and the Guangxi Natural Science Foundation (No.2018GXNSFAA294008, 2018GXNSFDA281004).”

“We thank the subsidization of the National Natural Science Foundation of China (No.31971664, 31560212) and the Guangxi Natural Science Foundation (No.2018GXNSFAA294008, 2018GXNSFDA281004).”

4. Please include your tables as part of your main manuscript and remove the individual files. Please note that supplementary tables (should remain/ be uploaded) as separate ""supporting information"" files

Reviewers' comments:

Reviewer's Responses to Questions

**Comments to the Author**

1. Is the manuscript technically sound, and do the data support the conclusions?

Reviewer #1: Partly

Reviewer #2: Yes

2. Has the statistical analysis been performed appropriately and rigorously? 

Reviewer #1: Yes

Reviewer #2: Yes

3. Have the authors made all data underlying the findings in their manuscript fully available?

Reviewer #1: Yes

Reviewer #2: Yes

4. Is the manuscript presented in an intelligible fashion and written in standard English?

Reviewer #1: Yes

Reviewer #2: No

5. Review Comments to the Author

Reviewer #1: The study and manuscript by Peng et al. are straight-forward, clear and interesting.

Since HAP1 and Hap2 are closely related to each other and the authors suspect that the genetic diversity of linage A may be due to sexual reproduction between linages A and B I think that the study would benefit from further analysis of L. invasa males.

A representative number (N>20) of male samples from each linage should be added to the basin analysis in order to supply a better understanding on the genetic drift of L. invasa.

Reviewer #2: The paper analyzed the genetic diversity of Leptocybe invasa from china. I believe the data of is especially valuable and helpful for research community. However, there are significant corrections which need to be made with regards to the write up, grammar and syntax need to be corrected for readability as the manuscript results are valuable to a diverse range of readers. Furthermore, the figures and tables need to be self-explanatory and thus the description of the figures and tables in the captions need to be revised.

Kindly find my revisions to the grammar within the attached document.

6. PLOS authors have the option to publish the peer review history of their article (what does this mean?). If published, this will include your full peer review and any attached files.

Reviewer #1: No

Reviewer #2: No

---

## [Author Response · Author response to Decision Letter 0]

23 Sep 2021

Description of manuscript modifications

PLoS One editorial department:

 First, we would like to thank the editors and reviewers for their hard work. The reviewers provided very good modification opinions for the manuscript. The explanations of the revisions made based on the modification opinions put forward by the academic editors and experts follow.

Recommendations of Academic editors:

 Please modify and supplement the manuscript according to the format requirements of PLoS One.

Author's response to Academic editors:

 The fund related information has been deleted from the manuscript, the ethics statement has been added to the material and methods, the tables has been added to the corresponding position of the manuscript as required, the figures have been uploaded in TIF format as required, and the format of the reference literature has been modified as required.

Recommendations of Reviewer 1:

 The study and manuscript by Peng et al. are straight-forward, clear and interesting.Since HAP1 and Hap2 are closely related to each other and the authors suspect that the genetic diversity of linage A may be due to sexual reproduction between linages A and B I think that the study would benefit from further analysis of L. invasa males.A representative number (N>20) of male samples from each linage should be added to the basin analysis in order to supply a better understanding on the genetic drift of L. invasa.

Author's response to Reviewer 1:

 Adding male samples to each population is indeed of great significance to explain the genetic drift of L. invasa. Especially in populations with introgressive hybridization, the male genotype will be strong evidence to verify the introgressive hybridization between lineages. Unfortunately, parthenogenesis is still the main reproductive mode of L. invasa, and it is difficult for us to collect samples of male adults in the field. For example, in the nearly 300 L. invasa samples collected from the GXFCG1 population, only 2 were males, while in the nearly 200 L. invasa samples collected from the GXFCG2 population, only 4 were males. Moreover, as we did not collect males from most geographical populations, we had too few male samples for a population genetic analysis. At present, we may not be able to supplement the male samples of each population (> 20), but we are also trying to collect a large number of male samples to supplement our experiments in the future. We will further monitor the introgressive hybridization of L. invasa to provide a reference for preventing the spread of invasive pests worldwide.

Recommendations of Reviewer 2:

 paper analyzed the genetic diversity of Leptocybe invasa from china. I believe the data of is especially valuable and helpful for research community. However, there are significant corrections which need to be made with regards to the write up, grammar and syntax need to be corrected for readability as the manuscript results are valuable to a diverse range of readers. Furthermore, the figures and tables need to be self-explanatory and thus the description of the figures and tables in the captions need to be revised. Kindly find my revisions to the grammar within the attached document.

Author's response to Reviewer 2:

 The table has been inserted after the first quoted paragraph, and the title of the figure has been inserted after the first quoted paragraph, according to the formatting requirements of journal. The notes for the tables and figures have been supplemented. Additionally, the full text of the manuscript was polished to ensure its English readability.

Amend the notes in the attached document to read as follows

Note 1: how many samples were obtained at each location. This should be stated here or in the table.

Answer: The number of samples collected by each geographical population has been supplemented in the sample size column in Table 1.

Note 2: you wrote it for the audience. Some people may not know what nanodrop is.

Answer: The mention of a nanodrop in this article has been changed to a Nanodrop 2000 spectrophotometer.

Note 3: This statement needs to be revised

Answer: The language expression has been modified, see lines 158-159 for details.

Note 4: This statement is confusing please re-write

Answer: The language expression has been modified, see lines 166-167 for details.

---

## [Editor Report · Decision Letter 1]

1 Oct 2021

Genetic diversity analysis of the invasive gall pest Leptocybe invasa (Hymenoptera: Apodemidae) from China

PONE-D-21-20406R1

Dear Dr. Yang,

We’re pleased to inform you that your manuscript has been judged scientifically suitable for publication and will be formally accepted for publication once it meets all outstanding technical requirements.

Kind regards,

Bi-Song Yue, Ph.D

Academic Editor

PLOS ONE

---

## [Editor Report · Acceptance letter]

6 Oct 2021

PONE-D-21-20406R1 

Genetic diversity analysis of the invasive gall pest *Leptocybe invasa* (Hymenoptera: Apodemidae) from China 

Dear Dr. Yang:

I'm pleased to inform you that your manuscript has been deemed suitable for publication in PLOS ONE. Congratulations! Your manuscript is now with our production department. 

Kind regards, 

on behalf of

Dr. Bi-Song Yue 

Academic Editor

PLOS ONE